# Dietary Behaviour Is Associated with Cardiometabolic and Psychological Risk Indicators in Female Hospital Nurses—A Post-Hoc, Cross-Sectional Study

**DOI:** 10.3390/nu11092054

**Published:** 2019-09-02

**Authors:** Tasuku Terada, Matheus Mistura, Heather Tulloch, Andrew Pipe, Jennifer Reed

**Affiliations:** 1Exercise Physiology and Cardiovascular Health Lab, Division of Cardiac Prevention and Rehabilitation, University of Ottawa Heart Institute, Ottawa, ON K1Y 4W7, Canada; 2Faculty of Medicine, University of Ottawa, Ottawa, ON K1N 6N5, Canada; 3School of Human Kinetics, Faculty of Health Sciences, University of Ottawa, Ottawa, ON K1N 6N5, Canada

**Keywords:** Profile of Mood States (POMS), obesity, nutrition, shift-work

## Abstract

Unfavourable dietary behaviours of female nurses, especially among shift-working nurses, including high snacking frequency, short fasting period and large day-to-day energy intake variability may be linked with adverse health. In this study we: (1) examined the relationship between dietary behaviour and cardiometabolic and psychological health in female nurses; and, (2) compared dietary behaviour, cardiometabolic and psychological health between shift-working and non-shift-working female nurses. A total of 73 nurses had their cardiometabolic health indicators evaluated and completed psychological health questionnaires; 55 completed a 3-day dietary log. Associations between dietary behaviour and health measures were examined using Spearman’s partial correlation analysis. Analysis of covariance (ANCOVA) was used to compare dietary behaviour and health indicators between shift- and non-shift-working nurses. The majority of snacks consumed by nurses (70%) were unhealthy snacks (e.g., chocolate and chips), and higher snacking frequency was associated with greater percent body fat (r(50) = 0.287, *p* = 0.039), and worse mood-tension (r(48) = 0.327, *p* = 0.021) and anger-hostility (r(48) = 0.289, *p* = 0.042) scores. Day-to-day energy intake variability was positively associated with body mass index (BMI, r(50) = 0.356, *p* = 0.010) and waist circumference (r(50) = 0.283, *p* = 0.042). Shift-working nurses exhibited shorter fasting duration, larger day-to-day energy intake variability and higher total mood disturbance score when compared to their non-shift-working colleagues (all *p* < 0.05). The results of the present study suggested that addressing dietary behaviours may improve the cardiometabolic and psychological health of female nurses. Shift-working nurses may require a more specific dietary program to improve their psychological health.

## 1. Introduction

Nurses constitute the largest group of healthcare professionals in most countries including Canada [1,2]. Nurses possess high health literacy; however, many face the same cardiovascular disease risks as the populations they serve. A National Survey of the Work and Health of Nurses in Canada (N = 18,676) revealed that an alarming proportion of Canadian nurses report cardiometabolic and psychological health risks, including overweight or obesity (45%); hypertension (13%); smoking (11%); dyslipidemia (10%); diabetes (3%); job strain (31%); and, depression (9%) [2]. Diminished cardiometabolic and psychological health not only contribute to a heightened risk of cardiovascular disease [2] but also high rates of absenteeism and presenteeism (i.e., being at work but not functioning at full capacity) [3], both of which negatively impact patient care. In Canada and the United States, 9% of nurses were absent each week in 2016 [4] and 62% reported some degree of presenteeism [5]. The overwhelming majority of Canadian nurses are female (94.5%) [2], and often face high cardiovascular [6] and psychological [7] health risks.

Diet plays a pivotal role in maintaining and improving cardiometabolic [8] and psychological [9] health. Poor dietary quality (i.e., low adherence to national dietary guidelines) promotes the development and progression of several cardiometabolic disorders [10] and predicts an elevated risk of depression [11] and reduced well-being [12]. Further, the emerging literature suggests that particular dietary behaviours are associated with reduced cardiometabolic and psychological health. High snacking frequencies, a diminished fasting period and large day-to-day energy intake variability have been linked with weight gain [13], increased percent body fat [14], obesity [15,16,17], central obesity [18], decreased stress resistance [19] and increased risk of psychiatric disorders [20].

The high occupational demands experienced by nurses increase the incidence of irregular dietary behaviour [21]. Many nurses report irregular number of meals [22], frequent snacking [15,23,24] and stress-induced eating [22]; all behaviours are more prominent during shift-work [22], a common practice within the nursing workforce [25,26]. When compared to non-shift-working nurses, shift-workers manifest higher cardiometabolic and psychological health risks, including higher body mass index (BMI), higher waist-to-hip ratio [27] and poorer mental health [28,29]. The risk of developing coronary heart disease is higher in shift-working nurses [30].

Despite the accumulation of evidence highlighting the important role of dietary behaviours, much research to date has focused predominantly on dietary quality; no study has examined the association of dietary behaviours with cardiometabolic and psychological health in female nurses. Moreover, no study has simultaneously compared dietary behaviour, cardiometabolic and psychological health between shift- and non-shift-working nurses. Understanding the link between dietary behaviour and cardiometabolic and psychological health may facilitate the development of recommendations and programs to enhance the health and well-being of female nurses.

The primary purpose of the current study was to examine whether dietary behaviour, such as snacking frequencies, fasting duration and day-to-day energy intake variability were associated with cardiometabolic and psychological health in female nurses. The secondary purpose was to compare dietary behaviours, and cardiometabolic and psychological health between shift- and non-shift-working female nurses. We hypothesized that higher snacking frequencies, shorter duration of the fasting state and larger day-to-day energy intake variability would be associated with findings of adverse cardiometabolic and psychological health. We also hypothesized that, when compared to non-shift-working female nurses, shift-working female nurses would report more frequent snacking, shorter fasting duration and greater day-to-day energy intake variability, and more findings of adverse cardiometabolic and psychological health.

## 2. Materials and Methods

### 2.1. Study Design

This is a secondary data analysis of a single-center randomized trial examining the impact of 6-week web-based feedback on physical activity levels of nurses [4]. The study was conducted at the University of Ottawa Heart Institute (UOHI), a tertiary care cardiovascular institute.

### 2.2. Participants

The recruitment strategies have been described previously [4]. Briefly, a convenience sample of participants was recruited between September and November 2013. Eligible participants were: (1) registered nurses; (2) able to walk unassisted; and, (3) willing to wear a physical activity monitoring device. We excluded those who were: pregnant or lactating; unable to read and understand English; or had medical contraindications to exercise. Nurses were categorized as ’shift-working’ or ’non-shift-working’. We defined shift-work as ’work outside of daytime hours including irregular or rotating schedules, evening and night work’ [26]. All participants provided written consent in accordance with the Declaration of Helsinki. The protocol was approved by the UOHI Human Ethics Board (Protocol No. 20130429).

### 2.3. Dietary Behaviour and Intake

Dietary behaviour, total energy and macronutrient intake were assessed using 3-day dietary logs. The 3-day dietary log has been shown to produce a high degree of accuracy in reporting food intake when compared with direct observation [31]. A member of the research team with training in nutrition instructed participants to record all consumed foods and beverages. We provided participants with a food portion package, which contained diagrams illustrating container sizes, cuts of meat, and various geometric figures that were used to estimate portion sizes for foods. Participants were instructed to use the scale diagrams as a tool for describing food intake. Participants recorded all foods and beverages consumed, along with time of day, location and meal type (e.g., breakfast). A sample page of an accurately completed dietary record was appended to the dietary log as a reference.

All food and beverages consumed within a 30 min period in the same location were included as the same eating occasion [32,33]. Energy intake consisting of only a beverage containing <10 kcal (e.g., diet drinks or coffee without cream or sugar) was not counted as a snack. Dietary intake data were analyzed using the Nutritionist Pro Diet Analysis software v.4.5 (Axxya Systems, Stafford, TX, USA). Data were entered into the software by research assistants and subsequently verified by TT and MM. When errors were identified, changes were made following agreement between TT and MM. For each participant, average total daily energy and macronutrient (i.e., carbohydrate, protein and fat) intake, average daily snacking frequency, longest fasting duration and day-to-day energy intake variability were determined. Snacks were categorized as being healthy (e.g., fruits, vegetables, yogurt and nuts) or unhealthy snacks (e.g., chips, sweets and chocolates). The longest fasting duration was defined as the longest period without any energy consumption between the first meal/snack on day 1 and the last meal/snack on day 3. Day-to-day energy intake variability was determined using coefficient of variation (CV), which is the standard deviation divided by the mean multiplied by 100 [34]. We only analyzed dietary logs completed at baseline because the intervention of the original trial did not target dietary changes. Further, only half of the nurses who completed the baseline dietary logs also completed dietary logs following the intervention.

### 2.4. Cardiometabolic Health Indicators

Cardiometabolic health indicators included: body mass; BMI; waist circumference; percent body fat; resting blood pressure (BP); and, resting heart rate (HR). Cardiometabolic health measures were taken between 06:30 and 10:00 hours. Height was measured to the nearest 0.1 cm, body mass was measured to the nearest 0.1 kg, and BMI was calculated (kg/m^2^). Waist circumference was measured to the nearest 0.5 cm (Seca201) at the narrowest point of the torso while participants stood with arms at their sides, feet together and abdomen relaxed [35]. Body fat percentage was measured using bioelectrical impedance (BIA) (UM-041, Tanita, Roxton Industries Inc., Kitchener, Ontario). Participants were asked to adhere to the following prior to their anthropometric measurements: (1) no eating or drinking for 4 h; (2) no moderate- or vigorous-intensity physical activity for 12 h; (3) no alcohol consumption for 48 h; (4) to empty their bladder within 30 min; (5) to refrain from consuming caffeine and diuretic use unless prescribed by a physician; and, (6) to postpone measurements if retaining water due to changes in menstrual cyclicity. Resting BP and HR were assessed in a seated position after a 5 min rest period using an automated, non-invasive BP monitor (Bp-TRU, Coquitlam, BC, Canada). All measurements were performed in triplicate and the average was reported for descriptive purposes.

### 2.5. Psychological Health Indicators

Psychological measures included the Profile of Mood States (POMS) and Eating Disorder Inventory-3 (EDI-3) questionnaires. The POMS consists of 65 items with 5-point subjective scales evaluating 7 aspects of mood: tension–anxiety; depression–dejection; anger–hostility; confusion–bewilderment; vigor–activity; fatigue–inertia; and, confusion–bewilderment, as well as their composite summary score, the total mood disturbance (TMD) score. The POMS has undergone rigorous psychometric testing and has demonstrated excellent reliability as well as content, construct, and criterion validity [36]. High scores in tension–anxiety, depression–dejection; anger–hostility; confusion–bewilderment; fatigue–inertia; confusion–bewilderment; TMD; and, lower score in vigor–activity indicate higher psychological distress.

The EDI-3 is a self-reported questionnaire used to measure psychological traits known to be closely related to eating disorders. The EDI-3 consists of 91 items organized into 3 eating-disorder-specific scales: drive for thinness; bulimia; and, body dissatisfaction, and 9 general psychological scales that are highly relevant to, but not specific to, eating disorders: low self-esteem; personal alienation; interpersonal insecurity; interpersonal alienation; interoceptive deficits; emotional dysregulation; perfectionism; asceticism; and, maturity fears [37]. Higher scores are related with greater eating and psychological disorders [37]. The EDI-3 has demonstrated good internal consistency and discriminative validity, and excellent sensitivity and specificity [38].

### 2.6. Data Analysis

Statistical analysis was performed using SPSS version 24 (IBM Corp, Armonk, NY, USA). Demographic data are presented as mean ± standard deviation (SD). Spearman’s partial correlation analysis adjusted for age was used to evaluate the relationship between total energy intake and macronutrient intake with cardiometabolic and psychological health indicators. To examine associations of snacking frequency, fasting duration and day-to-day energy intake variability with cardiometabolic and psychological health indicators, we used the Spearman’s partial correlation analysis while adjusting for age and mean daily energy intake. Analysis of covariance (ANCOVA) while adjusting for age was used to compare energy intake, macronutrient intake, dietary behaviour, cardiometabolic and psychological health indicators between shift- and non-shift-working nurses. The level of significance was set at *p* < 0.05.

## 3. Results

A total of 73 female nurses were included in the study. Nurses’ demographics and cardiometabolic and psychological health indicators according to work types (i.e., shift- vs. non-shift-working) are presented in Table 1. Of the 73 female nurses included in the study, 55 completed a 3-day dietary log. There were no differences in cardiometabolic or psychological health indicators between nurses who completed the dietary log and those who did not. The associations of total energy intake, macronutrient intake and dietary behaviour with cardiometabolic and psychological health indicators are summarized in Table 2. There were no significant correlations between the predictor variables (i.e., snacking frequency, fasting duration and day-to-day energy intake variability).

There were no significant associations between energy intake and cardiometabolic health indicators or between macronutrient intake and cardiometabolic health indicators. Dietary behaviour was positively associated with cardiometabolic health risks. Snacking frequency was positively associated with percent body fat (r(50) = 0.287, *p* = 0.039); day-to-day energy intake variability was positively associated with BMI (r(50) = 0.356, *p* = 0.010) and waist circumference (r(50) = 0.283, *p* = 0.042, Appendix A). The majority of snacks (70% of overall snacks) were unhealthy snacks, such as chocolate, chips and sweetened beverages, whereas only 30% of overall snacks were heathy snacks (e.g., fruits, vegetables, yogurt and nuts).

Higher energy intake and carbohydrate intake were associated with worse tension–anxiety, depression–dejection, fatigue–inertia, vigor–activity and TMD scores (all *p* < 0.05, Table 2). Higher energy intake and fat intake were also inversely associated with low self-esteem, personal alienation and interoceptive deficits scores (all *p* < 0.05, Table 2). Snacking frequency was associated with worse tension–anxiety (r(48) = 0.327, *p* = 0.021) and anger–hostility (r(48) = 0.289, *p* = 0.042) scores. There were no significant associations between dietary behaviour and psychological traits related to eating disorders measured by the EDI-3.

Shift-working nurses were significantly younger when compared to non-shift-working nurses (41.3 ± 11.9 vs. 50.7 ± 7.8 years, *p* < 0.001, Table 1). When compared to non-shift-working nurses, shift-working nurses showed significantly shorter fasting duration (796.3 ± 112.8 vs. 707.7 ± 120.6 min, *p* = 0.015) and larger day-to-day energy intake variability (13.8 ± 7.5 vs. 22.8 ± 10.2%, *p* = 0.002). Among shift-working nurses, overall energy intake did not differ between shift-working and non-shift-working days (2083 ± 872 vs. 1907 ± 617 kcal/day, *p* = 0.321). We found that 61.5% shift-working nurses had energy intake discrepancy >400 kcal/day between shift-working and non-shift-working days. Indicators of psychological health, including anger–hostility, fatigue–inertia, confusion–bewilderment and TMD scores were significantly worse in shift-working nurses when compared to non-shift-working nurses (all *p* < 0.05, Table 1). There were no differences in energy intake, macronutrient intake or any of the cardiometabolic health indicators between shift- and non-shift-working nurses.

## 4. Discussion

Studies of the nutritional practices of nurses to date have primarily focused on what they eat; less attention has been given to how they eat [39] and associations with cardiometabolic and psychological health. The principal finding of the current study was that dietary behaviour was related with the cardiometabolic and psychological health of female nurses. Higher snacking frequency was associated with increased risk of obesity and central obesity, as well as higher tension–anxiety and anger–hostility. Larger day-to-day energy intake variability was also linked with increased risk of obesity and central obesity. When compared to non-shift-working nurses, shift-working nurses exhibited significantly shorter fasting duration and larger day-to-day energy intake variability and several greater psychological distresses, such as higher anger–hostility, fatigue–inertia, confusion–bewilderment and TMD scores.

The results of this study showed that neither total energy intake nor macronutrient intake but rather dietary behaviour (i.e., higher snacking frequency and large day-to-day energy intake variability) was significantly correlated with cardiometabolic health indicators, including BMI, percent body fat and waist circumference. Considering that 40% of nurses who are overweight or obese report difficulty controlling weight while eating a healthy diet [40], the results of this study suggest that dietary behaviour modifications may be a strategic target to manage the high prevalence of overweight, obesity and related cardiovascular disease [6] among female nurses. It is not clear how day-to-day energy intake variability was linked to BMI and waist circumference. However, if the large variability was arising from additional energy intake on one of the three measurement days, such extra calories might have resulted in a positive energy balance and contributed to a greater BMI and waist circumference. Alternatively, it is also possible that flexibility in energy intake may create more opportunities for loss of dietary control and contribute to higher body mass [13]. The results of this study indicating significant association of day-to-day energy intake variability with BMI and waist circumference are similar to those of previous studies linking energy intake variability with percent body fat in Caucasians living in the Czech Republic [14] and with greater weight gain among individuals in a National Weight Control Registry database [13].

In addition to day-to-day energy intake variability, snacking frequency was associated with higher percent body fat. Studies examining the relationship between snacking and obesity to date have shown mixed results, with some reporting positive [41] while others reporting inverse associations between snacking frequency and overweight and abdominal obesity [42]. The discrepancy is predominantly attributable to snack choice, with heathy snacks (e.g., fruits, vegetables, yogurt and nuts) linked to less adiposity while unhealthy snacks (e.g., chips, sweets and chocolates) are linked to high adiposity [41]. Because eating a small amount of sweet food, such as chocolate ephemerally improves negative mood state [43,44], it is possible that unhealthy snacking to cope with stress was linked to greater percent body fat in female nurses. When we assessed the snack quality of female nurses in the current study, we found that 70% of overall snacks were unhealthy (e.g., chocolate, potato chips and sweetened beverages). Reducing unhealthy snacks while increasing healthy snacks may improve overweight and obesity among female nurses.

In addition to poor cardiometabolic health, diminished psychological health is another important predictor of cardiovascular disease [45]. The mood scores measured by POMS in this study were comparable to those reported by generally healthy menopausal women [46] but slightly lower possibly due to younger age (53 ± 5 vs. 46 ± 10 years) [47]. The results of the present study showed that higher energy intake and carbohydrate intake were positively associated with tension–anxiety, depression–dejection, fatigue–inertia and TMD, while inversely associated with vigor–activity. As noted earlier, in individuals experiencing high psychological stress [2], food reward systems promote consumption of carbohydrate to reduce stress [48]. Given the significant association between higher snacking frequency and elevated mood–tension and anger–hostility, we speculate that female nurses under greater psychological distress had higher energy and carbohydrate intake as a result of increasing the number of unhealthy snacks. Similar to the relationship between obesity and snacking, a previous study with more than 800 nurses showed that unhealthy snacking is associated with higher psychological stress [49], whereas consumption of fruit is associated with lower anxiety, depression, fatigue, emotional eating and distress [50]. Switching from unhealthy snacks to healthy snacks may ameliorate psychological disturbance among female nurses.

With regard to the psychological health measured by the EDI-3 questionnaire, none of the predictor variables included in the present study were associated with eating-disorder-specific scales (i.e., drive for thinness, bulimia and body dissatisfaction). However, the amount of energy intake and fat intake were inversely associated with some general psychological scales that are relevant to eating disorders: low self-esteem, personal alienation and interoceptive deficit. We suspect that psychological vulnerability of female nurses resulted in dietary restraint. The negative correlations between energy intake and EDI-3 scores observed in the present study are similar to those of Iorga et al., who showed negative associations between several EDI-3 subscale scores and a tendency to skip meals and restrict energy intake in predominantly (>90%) female pharmacy students [51]. The EDI-3 subscale scores of the current study were also similar to those reported by Iorga et al. [51].

Between shift- and non-shift-working female nurses, we found that shift-working nurses had significantly shorter fasting duration and greater day-to-day energy intake variability. These findings are consistent with those of a previous study indicating associations of shift-work with a more unbalanced temporal eating pattern and diet [52]. Because shift-work redistributes food intake from day to night [53], decreased duration of overnight fasting may have resulted in shorter fasting duration in shift-working nurses. Additionally, shift-working nurses often have limited food accessibility and healthy options in the evening and at night [24,54]; they consume more sweet food during night-duty in order to maintain wakefulness and energy levels [55]. Poor sleep quality and short sleep duration during shift-work may also have led to an increased feeling of hunger [56]. A relatively large proportion of shift-working nurses showing high energy intake discrepancy between shift-working and non-shift-working days (>400 kcal) may have contributed to larger day-to-day energy intake variability among shift-working nurses when compared to non-shift-working nurses.

The identification of higher psychological distress in shift-working nurses when compared to non-shift-working colleagues after adjusting for age confirmed the finding from a previous study linking shift-work and poor mental health in nurses [28]. The results of the present study suggest that shift-working female nurses are in greater need of supportive psychological health strategies when compared to non-shift-working colleagues. Given low physical activity levels, especially in shift-working nurses [57], and high sedentary behaviour [57,58], strategies to implement a more active lifestyle in addition to healthy dietary behaviours may play an important role in improving the health of female nurses. Psychological supports and dietary counseling may also assist nurses who are suffering from high psychological distress.

There are limitations to our findings that must be acknowledged. First, the results of this study are based on post-hoc analyses. The primary objective of the original study was to examine changes in physical activity levels using an activity monitor [4]; examining the linkages between dietary behaviour and cardiometabolic and psychological health were a secondary objective. Wearing the activity monitor while completing dietary logs may have influenced the nurses’ dietary behaviour by increasing health awareness. Second, given the cross-sectional nature of this study, future longitudinal studies are needed to determine whether modifying dietary behaviour would improve cardiometabolic and psychological health of female nurses. Third, the health effects of food likely represent the synergy of composite effects and interactions of multiple factors. Residual confounders not included in our analyses, such as energy expenditure, cannot be neglected. Fourth, self-reported dietary behaviours are subject to a social desirability bias. While dietary logs in which participants record the timing and food items consumed shortly after the meal are demonstrated to be a reliable and valid tool [59], it is possible that female nurses cognizant of the study altered their dietary behavior.

## 5. Conclusions

This study identified dietary behaviour was linked with cardiometabolic and psychological health in female nurses. The findings suggest that frequent snacking and large day-to-day energy intake variability are linked with higher cardiometabolic and psychological risks in female nurses. Nurses work in highly stressful environments and, despite high medical literacy, they are not protected from cardiovascular disease. This emphasizes the need for strategies to implement healthy dietary behaviours to improve their cardiometabolic and psychological health, particularly among those who are involved in shift-work. Strategies targeting dietary behavior have a potential to improve the cardiometabolic and psychological health of female nurses.

## Figures and Tables

**Table 1 nutrients-11-02054-t001:** Dietary patterns, cardiometabolic and psychological health of female shift-working and non-shift-working nurses.

	All (N = 73)	Non-Shift- Working (n = 40)	Shift-Working (n = 32)	Non-Shift vs. Shift *p* Value
Age, year	46.6 (10.8)	50.7 (7.8)	41.3 (11.9)	**<0.001**
Height, cm	165.0 (6.1)	164.6 (5.8)	165.6 (6.6)	0.395
Body mass, kg	75.4 (16.4)	72.5 (14.4)	79.3 (18.3)	0.130
BMI, kg/m^2^	27.7 (5.6)	26.7 (5.1)	28.9 (6.0)	0.059
Percent body fat, %	37.2 (7.7)	36.3 (7.1)	38.5 (8.5)	0.097
Waist circumference, cm	84.2 (12.6)	83.1 (12.0)	85.5 (13.6)	0.119
SBP, mmHg	114 (12)	116 (15)	113 (9)	0.977
DBP, mmHg	75 (7)	75 (9)	74 (6)	0.304
Heart rate, bpm	68 (9)	67 (8)	68 (10)	0.932
Physical activity, steps/day	9556 (2285)	9232 (1636)	9975 (2896)	0.740
Diet	N = 55	n = 34	n = 21	
Caloric intake, kcal/day	1793.6 (558.1)	1639.2 (497.3)	2043.7 (571.7)	**0.011**
Caloric intake per body mass, kcal/day/kg	24.5 (8.1)	23.4 (8.3)	26.3 (7.8)	0.104
Protein intake, g/day	81.6 (28.2)	77.1 (23.2)	88.9 (34.2)	0.066
Protein intake per body mass, g/day/kg	1.1 (0.5)	1.1 (0.4)	1.2 (0.5)	0.217
Carbohydrate intake, g/day	194.9 (74.7)	174.4 (61.4)	228.1 (83.5)	**0.020**
Carbohydrate intake per body mass, g/day/kg	2.7 (1.0)	2.5 (0.9)	2.9 (1.2)	0.113
Fat intake, g/day	71.2 (26.6)	65.5 (26.4)	80.5 (24.7)	0.058
Fat intake per body mass, g/day/kg	1.0 (0.4)	0.9 (0.4)	1.0 (0.3)	0.317
Snacking frequency, n/day	1.8 (1.0)	1.8 (1.1)	1.9 (0.9)	0.785
Fasting duration, min/day	768.9 (121.1)	796.3 (112.8)	707.7 (120.6)	**0.015**
Day-to-day energy intake variability, %	17.3 (9.6)	13.8 (7.5)	22.8 (10.2)	**0.002**
POMS	N = 69	n = 38	n = 31	
Tension–anxiety	7.5 (5.8)	6.7 (4.5)	8.5 (7.0)	0.217
Depression–dejection	6.4 (7.7)	4.8 (5.6)	8.5 (9.4)	**0.045**
Anger–hostility	6.1 (6.8)	4.5 (5.2)	8.0 (8.0)	**0.011**
Vigor–activity	15.1 (5.6)	15.8 (5.9)	14.2 (5.2)	0.081
Fatigue–inertia	7.8 (5.5)	6.4 (4.8)	9.5 (5.9)	**0.012**
Confusion–bewilderment	4.9 (4.1)	4.4 (3.5)	5.6 (4.6)	0.382
Total mood disturbance	17.7 (30.3)	11.0 (24.2)	25.9 (35.2)	**0.027**
EDI-3	N = 68	n = 37	n = 31	
Drive for thinness	7.9 (2.8)	8.0 (2.9)	7.8 (2.8)	0.551
Bulimia	6.4 (3.3)	5.6 (2.8)	7.3 (3.6)	0.260
Body dissatisfaction	9.7 (6.4)	10.2 (6.1)	8.9 (6.8)	0.704
Low self-esteem	9.0 (1.8)	9.0 (1.7)	9.0 (1.9)	0.816
Personal alienation	10.3 (2.3)	10.5 (2.0)	9.9 (2.5)	0.209
Interpersonal insecurity	16.6 (4.3)	17.3 (3.8)	15.6 (4.8)	0.240
Interpersonal alienation	15.4 (3.4)	16.2 (2.5)	14.5 (4.1)	0.100
Interoceptive deficits	8.8 (2.9)	8.5 (2.9)	9.2 (0.8)	0.755
Emotional dysregulation	4.9 (2.4)	4.9 (2.4)	4.9 (2.4)	0.906
Perfectionism	4.7 (2.6)	4.9 (2.4)	4.4 (2.8)	0.533
Asceticism	5.0 (3.1)	4.7 (2.8)	5.4 (3.5)	0.545
Maturity fears	13.0 (3.2)	13.4 (3.4)	12.6 (3.0)	0.788

Data are presented as mean (SD). BMI: body mass index; CV: coefficient of variation; DBP: diastolic blood pressure; EDI-3: Eating Disorder Inventory 3; POMS: The Profile of Mood States; SBP: systolic blood pressure. One participant who completed cardiometabolic measures had work shift information missing. This resulted in a mismatch between the total number of nurses and the sum of shift-working and non-shift-working nurses for cardiometabolic measures.

**Table 2 nutrients-11-02054-t002:** Correlations between dietary intake and cardiometabolic and psychological health indicators.

	Daily Energy Intake, Kcal/Day	Daily Protein Intake, g/Day	Daily Carbohydrate Intake, g/Day	Daily Fat Intake, g/Day	Day-to-Day Caloric Intake Variability, CV	Snacking Frequency, n/Day	Longest Fasting Period, min
Cardiometabolic health indicators							
BMI, kg/m^2^	0.215 (0.119)	0.235 (0.087)	0.125 (0.368)	0.228 (0.097)	**0.356** **(0.010)**	0.098 (0.492)	−0.010 (0.951)
Percent body fat, %	0.218 (0.113)	0.200 (0.147)	0.174 (0.208)	0.124 (0.370)	0.256 (0.067)	**0.287** **(0.039)**	−0.074 (0.650)
WC, cm	0.175 (0.207)	0.163 (0.240)	0.105 (0.451)	0.137 (0.323)	**0.283** **(0.042)**	0.085 (0.549)	−0.064 (0.693)
SBP, mmHg	0.195 (0.158)	0.083 (0.549)	0.251 (0.067)	0.052 (0.710)	−0.089 (0.583)	0.045 (0.754)	−0.272 (0.090)
DBP, mmHg	0.153 (0.269)	0.109 (0.431)	0.166 (0.230)	0.082 (0.553)	−0.216 (0.124)	−0.003 (0.985)	−0.213 (0.187)
HR, bpm	0.158 (0.251)	0.264 (0.052)	0.064 (0.643)	0.129 (0.349)	0.009 (0.947)	0.028 (0.859)	0.003 (0.984)
Psychological health indicators—POMS							
Mood/tension	**0.301** **(0.030)**	0.069 (0.626)	**0.395** **(0.004)**	0.158 (0.265)	0.197 (0.171)	**0.327** **(0.021)**	−0.028 (0.871)
Depression/dejection	**0.334** **(0.015)**	0.106 (0.456)	**0.377** **(0.006)**	0.234 (0.095)	0.262 (0.066)	0.265 (0.063)	−0.018 (0.918)
Anger/hostility	**0.274** **(0.049)**	−0.002 (0.991)	0.274 (0.050)	0.237 (0.091)	0.274 (0.054)	**0.289** **(0.042)**	0.025 (0.887)
Confusion/bewilderment	0.206 (0.144)	0.009 (0.949)	−0.232 (0.098)	0.139 (0.327)	0.160 (0.268)	0.174 (0.226)	0.120 (0.491)
Vigor/activity	**−0.328** **(0.017)**	−0.243 (0.082)	**−0.436** **(0.001)**	−0.037 (0.794)	−0.096 (0.508)	−0.050 (0.732)	−0.163 (0.349)
Fatigue/inertia	**0.417** **(0.002)**	**0.335** **(0.015)**	**0.500** **(<0.001)**	0.167 (0.236)	0.130 (0.369)	0.149 (0.303)	−0.156 (0.370)
TMD	**0.363** **(0.008)**	0.145 (0.307)	**0.429** **(0.002)**	0.197 (0.162)	0.231 (0.107)	0.257 (0.071)	0.033 (0.852)
Psychological health indicators—EDI-3							
Drive for thinness	−0.119 (0.401)	−0.084 (0.552)	0.000 (0.999)	−0.178 (0.208)	−0.080 (0.576)	−0.204 (0.150)	−0.030 (0.855)
Bulimia	−0.183 (0.194)	−0.068 (0.632)	−0.116 (0.411)	−0.184 (0.190)	−0.126 (0.377)	−0.034 (0.811)	0.030 (0.853)
Body dissatisfaction	−0.218 (0.121)	−0.189 (0.179)	−0.099 (0.483)	−0.210 (0.136)	−0.204 (0.152)	−0.209 (0.141)	−0.180 (0.266)
Low self-esteem	**−0.313** **(0.024)**	−0.157 (0.266)	−0.272 (0.051)	−0.211 (0.133)	0.011 (0.942)	−0.121 (0.399)	−0.045 (0.782)
Personal alienation	**−0.314** **(0.023)**	−0.150 (0.290)	−0.263 (0.060)	**−0.283** **(0.042)**	0.044 (0.758)	−0.143 (0.317)	0.008 (0.960)
Interpersonal insecurity	−0.250 (0.074)	−0.112 (0.427)	−0.244 (0.082)	−0.203 (0.149)	−0.027 (0.852)	−0.068 (0.634)	−0.112 (0.492)
Interpersonal alienation	−0.237 (0.090)	−0.029 (0.837)	−0.139 (0.327)	−0.252 (0.072)	−0.070 (0.625)	−0.089 (0.535)	−0.143 (0.378)
Interoceptive deficits	**−0.352** **(0.011)**	−0.248 (0.077)	**−279** **(0.045)**	**−0.292** **(0.036)**	−0.002 (0.989)	−0.125 (0.381)	−0.025 (0.876)
Emotional dysregulation	−0.177 (0.209)	−0.044 (0.759)	−0.101 (0.475)	−0.181 (0.199)	−0.008 (0.955)	−0.077 (0.592)	−0.167 (0.302)
Perfectionism	0.024 (0.865)	0.066 (0.644)	0.131 (0.354)	−0.042 (0.769)	0.107 (0.454)	0.010 (0.946)	0.080 (0.624)
Asceticism	−0.129 (0.363)	−0.103 (0.468)	−0.022 (0.877)	−0.232 (0.097)	−0.103 (0.474)	−0.069 (0.630)	−0.110 (0.499)
Maturity fears	−0.058 (0.683)	−0.078 (0.581)	−0.060 (0.675)	−0.033 (0.819)	0.000 (0.998)	−0.093 (0.518)	−0.241 (0.134)

Data are presented as Spearman’s partial correlation coefficients (*p*-value) controlling for age for mean caloric and macronutrient intake, and controlling for age and mean caloric intake for day-to-day caloric intake variability, snacking frequency and fasting periods. BMI: body mass index; DBP: CV: coefficient of variation; DBP: diastolic blood pressure; EDI-3: Eating Disorder Inventory-3; HR: heart rate; POMS: The Profile of Mood States; SBP: systolic blood pressure; TMD: total mood disturbance; WC: waist circumference.

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
