# Peer review of "Dietary Behaviour Is Associated with Cardiometabolic and Psychological Risk Indicators in Female Hospital Nurses—A Post-Hoc, Cross-Sectional Study"

_nutrients, 2019, doi:10.3390/nu11092054_

Round 1

Reviewer 1 Report

Overall evaluation: Overall, well-written essay investigating and identifying essential issues that has wide implications. The authors has to better acknowledge that the study results are based on a post-hoc analyses and that the original study design not was intended for this study and hence – that there may have been a lack of variables or imprecise variables in the study protocol. The study protocol was set up to answer questions on physical activity. To have an activity monitoring device applied while studying/analyzing dietary behaviour is associated with bias and this has to be mentioned in the limitations of the study section.

Specific comments:

Title – I suggest to add ”….female hospital nurses – a post-hoc cross-sectional study” Results - Line 193. ”However”, should be deleted in Results – it should only be used when interpreting data. Rephrase the sentence. Discussion – line 27, 229, 233, 254, 260, 273, 308. The ”present study” results – not ”our” results or findings. Discussion – line 254. This important data on snack quality should be inserted in Results as well and also in the Abstract. Discussion – line 291-suspicion of excess energy intake during shift work. Can that be demonstrated by inserting this data in Results? That would have been a great improvement of data. If not possible, acknowledge that and revise in the manuscript. Discussion -last sentence in the limitations of the study paragraph – line 303. Suggest to add ”longitudinal” to ”future longitudinal studies…” Discussion -limitations of the study section. Please insert a line that the study results are based on a post-hoc analyses and that the original study design was set up to answer other questions, namely on physical activity. See under ”overall evaluation”. Discussion – limitations of the study. For the purpose of this study – ”nurses planning to become pregnant” was not included. Is that scientifically sound? Those nurses should be a great part of the nurse working force. Please explain in the limitations of the study section. Conclusions. – line 310 – delete referencing in Conclusions. Discussion – the reader would like the authors to write a few lines in a separate paragraph with a few references on potential strategies to improve cardiometabolic and psychological health, particularly for shift-working nurses.

Author Response

We would like to thank the reviewer for the insightful comments and detailed suggestions. We have provided responses to each of the comments and carefully taken the comments into consideration in preparation of our revised manuscript. The attached document summarizes our responses to each comment.

Reviewer 2 Report

The manuscript by Terada et al. studies dietary behaviours of female nurses and their association with cardiometabolic and psychological risks. The study is well designed and results are well presented. Some minor comments.

Shift working nurses are significantly younger than non-shift working nurses. While adjustement for this counfounding variable was done for the assessment of energy intake with cardiometabolic and psychological health indicators, it is not clear if this was also done when assessing association between snacking frequency with health indicators. Please confirm. Some of the results are quite interesting and should be shown in a figure with regression lines, for example the association of snacking frequency with percent body fat, and the association of energy intake with BMI. Authors utilise ANCOVA for analysis which is appropriate; however considering the number of variables, did the authors consider a multilevel (hierarchical) modelling?

Author Response

We would like to thank the reviewer for the insightful comments and detailed suggestions. We have provided responses to each of the comments and carefully taken the comments into consideration in preparation of our revised manuscript. The attached file summarizes our responses to each comment. 

Round 2

Reviewer 1 Report

Good revision. One thing is left, the new inserted paragraph in Discussion can be improved with transfer of data to Results as well:

"We suspected that excess energy intake during shift-work would explain larger day-to-day energy intake variability in shift-working nurses and compared energy intake on shift-working and non-shift-working days. Overall energy intake was indeed greater on the shift-working days when compared to non-shift-working days; however, the difference was not statistically significant (1907 ± 617 vs. 2083 ± 872 kcal/day p=0.321). We also found that more than 60% of nurses had >400 kcal energy intake discrepancy between shift-working and non-shift-working days. Although it needs to be confirmed with a larger number of shift-working nurses, the large discrepancy in energy intake between shift-working and non-shift-working days may have contributed to large day-to-day energy intake variability."

This new results in Discussion has also to be placed in Results. Since, the p value was far from significant, there was no difference in energy intake so this has to be strictly written in Results. Please calculate and state the exact proportion of nurses who had > 400 kcal energy intake discrepancy in Results. When interpretation of results are done in Discussion, please delete mean + SD and p value. It was also confusing to see a higher energy intake for non-shift working days? Is that correct? The sentence is expressed erroneously and confusing. Please rephrase this paragraph.

Author Response

We would like to thank the reviewer for additional comments. Please see attached our responses to the comments. 
